# Monitoring and Controlling House Mouse, *Mus musculus domesticus*, Infestations in Low-Income Multi-Family Dwellings

**DOI:** 10.3390/ani11030648

**Published:** 2021-03-01

**Authors:** Shannon Sked, Salehe Abbar, Richard Cooper, Robert Corrigan, Xiaodan Pan, Sabita Ranabhat, Changlu Wang

**Affiliations:** 1Department of Entomology, Rutgers—The State University of New Jersey, 96 Lipman Dr., New Brunswick, NJ 08901, USA; Shannon.Sked@rutgers.edu (S.S.); abbar@sebs.rutgers.edu (S.A.); rcooper@sebs.rutgers.edu (R.C.); xiaodan.pan@rutgers.edu (X.P.); sabita.ranabhat12@gmail.com (S.R.); 2RMC Pest Management Consulting, LLC, Briarcliff Manor, NY 10510, USA; cityrats@mac.com

**Keywords:** *Mus musculus domesticus*, spatial distribution, rodent management, monitoring

## Abstract

**Simple Summary:**

The house mouse is a very common pest in low-income multi-family residential dwellings. They cause significant property damage and produce allergens that are linked to asthma and allergy. Current mouse management practices in these dwellings are not effective. This study attempted to gain insights into residents’ impressions of house mice, develop more effective mouse detection methods, and evaluate the effectiveness of building-wide mouse management programs. The programs were implemented by researchers for 63 days and the results were monitored for up to 12 months. Significant differences were found in the efficacy of two commercial blank baits for detecting house mouse activity. Chocolate spread was significantly more effective than both commercial blank baits for detecting house mice. Between the two commercial toxic rodent baits tested, FirstStrike^®^ (0.0025% difethialone) was more palatable than Contrac^®^ (0.005% bromadiolone) rodent bait. A building-wide mouse control program resulted in an 87% reduction in mouse activity after three months in two buildings. After 12 months, the number of infestations decreased by 94% in one building, but increased by 26% in another building. Long-term house mouse control requires continuous efforts and the incorporation of multiple strategies.

**Abstract:**

The house mouse, *Mus musculus domesticus*, is a common pest in multi-family residential apartment buildings. This study was designed to gain insights into residents’ impressions of house mice, develop more effective house mouse detection methods, and evaluate the effectiveness of building-wide house mouse management programs. Two high-rise apartment buildings in New Jersey were selected for this study during 2019–2020. Bait stations with three different non-toxic baits were used to detect house mouse activity. Two rodenticides (FirstStrike^®^—0.0025% difethialone and Contrac^®^—0.005% bromadiolone) were applied by researchers over a 63-day period and pest control operations were then returned to pest control contractors for rodent management. There were significant differences in the consumption rates of non-toxic baits and two toxic baits tested. A novel non-toxic bait, chocolate spread, was much more sensitive than the two commercial non-toxic baits for detecting mouse activity. The house mouse management programs resulted in an average 87% reduction in the number of infested apartments after three months. At 12 months, the number of infestations decreased by 94% in one building, but increased by 26% in the second building. Sustainable control of house mouse infestations requires the use of effective monitoring strategies and control programs coupled with preventative measures.

## 1. Introduction

The house mouse, *Mus musculus domesticus* Schwarz and Schwarz (Rodentia: Muridae), is a cosmopolitan rodent species. In the U.S., it is one of the most common indoor pests with significant public health implications [1,2,3,4,5,6]. The success of the house mouse is largely attributed to their ability to live in close association with humans [7]. Although effective rodenticides and trapping devices are available, mice remain a difficult pest to manage in multi-family dwellings (MFDs). Among 1363 persons surveyed in Baltimore, 49% of the respondents observed mice in their residences [8]. In Gary, Indiana, 36% of the surveyed apartments were infested [9]. In an urban housing community in Manchester, United Kingdom, 44% of the residents reported ongoing mouse infestations [10]. These numbers demonstrate the severity of mouse infestations in residential apartments, exemplifying house mice as important and prevalent urban pest species [11].

House mice also cause significant loss to human food through direct feeding and food contamination [11,12] and exploit human activities commensally [7,13]. Once established, they spread their territory through “budding” [14], which has been correlated to increases in population sizes with the spread of urbanization [15,16]. Their gnawing inflicts significant damage to buildings, infrastructures, and personal belongings [17]. Additionally, house mice transmit the infection agents of important diseases [3,5,6,18,19,20]. They are carriers and/or reservoirs of potential zoonotic viruses [5], pathogenic bacteria and antimicrobial-resistant genes [6]. The house mouse is also the source of mouse urinary protein allergens [21]. Mouse infestations in homes can result in rodent bites; throughout 1991–1994, 18.5% of the 514 New York residents surveyed experienced mouse bites [1]. For some individuals, mouse infestations evoke mental distress [22]. 

Rodent management has historically been challenging, requiring collaboration between property owners, the pest control industry, and local public health agencies [23]. The typical approach of managing house mouse infestation is reactionary, relying mainly on resident complaints and simple control tactics (glue traps and rodenticides) in individual apartments with little or no follow-up in many cases [24]. Residents often fail to report pest problems to property management either because they are unaware of the pest problem exists or are unwilling to report the pests [25]. An effective house mouse integrated pest management program (IPM) in MFDs should incorporate the education of residents, the elimination of rodent entry points into the building and within the building (exclusion), proper solid waste management (sanitation) to minimize food resource availability, the identification of all apartments with mouse activity, and active rodent control measures using U.S. Environmental Protection Agency registered baits and/or mechanical devices (snap traps, multi-catch traps, etc.). House mouse behavior and ecology can then be incorporated into the design of the treatment strategies. Previous studies have documented that house mouse territory was related to the distribution of resources and population dynamics [2,11,26,27,28]. Genetic studies indicated at the building level that house mouse populations were related [29]. But, there is a lack of knowledge regarding house mouse spatial distribution patterns in MFDs and a poor understanding of factors associated with their population structures.

The objectives of this study are: (1) gain insight into impressions of house mouse infestations in low-income high-rise MFDs among residents; (2) compare the effectiveness of various non-toxic food baits for detecting house mouse activity; (3) implement and evaluate building-wide house mouse monitoring and control strategies for multi-unit dwellings; and (4) assess the spatial distribution of house mice by floor elevation in high-rise apartment buildings.

## 2. Materials and Methods

### 2.1. Study Sites

Two low-income high-rise apartment complexes in New Jersey participated in the study. Both buildings are representative of the high-rise type models of low-income MFDs that began to dominate public housing in the 1950s [30]. One building in Trenton, New Jersey contained 246 apartment units on 15 total floors; each floor with a trash chute closet, along with office areas, storage rooms, a community room with a kitchen, a laundry room, a trash room, a compactor room, and a boiler room located on first floor and basement level. The second building, located in Linden, New Jersey, contained 200 apartments on 11 total floors along with office areas, a boiler room, a compactor room, laundry room, trash room and compactor room. The level of exclusion and sanitation of the common areas differed between the two buildings, with the Linden location having a higher level of sanitation in the trash compactor room and trash chute closets on each floor and a high level of exclusion along the building envelope perimeter. Sanitation and exclusion levels at Linden were rate 1 and were rated 3 at Trenton on a rating score of 1, 2 or 3 which represents “Good”, “Average” or “Poor”, respectively [31].

The two building are approximately 50 miles apart in two separate metropolitan areas within New Jersey, with Trenton being centrally located in the state and Linden being in the greater New York City metropolitan area. The buildings are typical of the constructions common for low-income high-rise developments in New Jersey. They were therefore comparable to each other while not being in the same neighborhood.

Occupants of the two buildings had a similar demographic distribution. The buildings were of similar size, style and construction, with both being brick and mortar on steel construction. In both buildings, each unit was a studio or one bedroom apartment. Two important differences existed between the Trenton and Linden buildings. First, the building in Trenton had an enclosed duct forced-hot-air heating system, whereas the building at Linden had baseboard hot water heating system. The baseboard hot water heating system included a hot water line that ran through walls between adjacent apartments and was not well sealed. Second, the building in Linden was built on a slab-on-grade construction style, while the Trenton building contained a basement. Both buildings are located in urban settings with grounds that included parking areas and manicured turf grass. There were minimal landscape plants or trees on the building grounds and neither had ornate landscaping immediately around the building façade at the ground level.

### 2.2. Initial Inspection and Resident Interview

At Linden and Trenton, 91% and 90%, respectively, of the apartments were accessed. Apartments that were not inspected were those with residents who refused the researchers access. Residents that refused the researchers access to their apartments were asked if they had observed house mice in their apartments. Vacant apartments were inspected as well.

To identify apartments containing mouse infestations, an initial cursory inspection was conducted on 23 January 2019 at the Trenton building and 26 February 2019 at the Linden building. The inspections included searching the kitchens, family rooms, bathrooms and bedrooms for evidence of mice, such as droppings or gnaw marks; questioning residents about whether they had experienced mice; and installing bait stations, which were left in place for one week. Protecta^®^ EVO^®^ Mouse bait stations (Bell Laboratories, Inc., Madison, WI, USA) were used to monitor rodent activity (Figure 1). The Protecta^®^ bait stations were baited with two commercially available non-toxic census (monitoring) food baits: Detex^®^ Soft Bait with Lumitrack^®^ (Bell Laboratories) and Liphatech^®^ Rat & Mouse Attractant™ (Liphatech, Inc. Milwaukee, WI, USA) along with three ~1 g dollops of Hershey’s Spreads Chocolate (The Hershey’s Company, Hershey, PA, USA). These baits are considered “non-toxic bait” in that they do not contain any rodenticide. All three bait choices were used in each station to eliminate the possible location effect when placed in different locations of an apartment.

Inside both buildings, two bait stations were placed per apartment: one at the baseboard adjacent to the stove in the kitchen and the second near the heating system. In the Trenton building, the second station was placed in the corner of the living room, along the baseboard, where the heating ductwork of a forced hot-air system was located. In the Linden building, the second station was placed in the corner of the living room, along the baseboard, where the hot water line entered the hydronic baseboard heating system.

In the non-apartment areas of the building, one station was installed in each trash-chute room on each floor and 2–6 stations were placed along the perimeter walls of the trash compactor room, laundry room and community rooms.

Residents that were home during the initial inspection were briefly interviewed in person in an effort to obtain resident beliefs and the actions taken to mitigate mouse issues. Questions included: (a) ”Do you see house mice in your apartment?”; (b) “What level of concern (very, concerned, not concerned) do you have about house mouse infestations?”; (c) and “What, if any, control products have you used to manage mice in your apartment?”; (d) “Do you agree or disagree that (1) mice cause disease, (2) mice easy to eliminate, and (3) mice prefer dirty apartments?”

One week after installing non-toxic bait stations, the stations were collected. The amount of consumption of Detex^®^ and Liphatech^®^ Rat & Mouse Attractant was measured by weighing the bait on a portable scale (Amir Digital Mini Scale, KA21-A-AUS, Shenzhen Amier Technology Co. Ltd., Shenzhen, China, 0.01–100 g) whenever there were observable signs of consumption such as bite marks or tearing at the bait package. Chocolate consumption was estimated by volume based upon visual observation. This method was selected based on findings from previous research, using the same bait and station design, showing that small chocolate dollops (~1 g), were fully consumed in each case by house mice. Therefore, the number of grams of chocolate spread consumed per station during data collection were recorded. An apartment was considered active for house mice if feeding on any of the commercial baits or chocolate spread in the two bait stations occurred.

All apartment and non-apartment areas of the building where mouse activity was found were further evaluated for conditions that are conducive to house mouse activity, including areas of clutter, sanitation issues, and entry points along the perimeter envelop of the building and interior walls between apartments and rooms. This was consolidated as a recommendations list of activities for the building management to conduct.

### 2.3. House Mouse Management: Design and Implementation

A management program to control the house mouse infestations in both buildings was designed and implemented. One week after collecting the non-toxic bait stations from the initial inspection, a recommendation list for exclusion and sanitation needs was given to the property managers based on observations made during the building-wide inspection.

Bait stations with rodenticides were installed in apartments and common areas where feeding occurred during the initial inspection. Bait stations were managed by the research team during the 63-day management program period. Three bait stations were placed in each apartment, two were placed in the same location used during the building-wide survey and the third was placed on the floor beneath the kitchen sink along the base of the wall. Bait stations were also placed along perimeter walls in the trash chute room in Trenton, since house mouse feeding occurred here during the initial inspections. Four stations were placed on either side of the two interior doors that led from the trash chute room to the hallway and an adjacent closet area and six stations were placed along the perimeter wall, spaced approximately 12 feet apart from each other. A total of 69 bait stations were installed at the Trenton location and 157 stations were installed at the Linden location.

The bait station type and layout of baits in the bait station was the same as it was for the initial inspection; however, the non-toxic baits were replaced with the counterpart bait packets that included rodenticide. Contrac^®^ Soft Bait (0.005% bromadiolone, Bell Laboratories) (15.29 g per bait packet) was used in place of Detex^®^ Soft Bait. FirstStrike^®^ Soft Bait (0.0025% difethialone, Liphatech, Inc.) (10.54 g per bait packet) was used in place of Liphatech^®^ Rat & Mouse Attractant. The active ingredients in both baits are second generation anticoagulants, which are commonly used by pest management professionals and constitute approximately 75% [32] to 90% [11] of the $784 million rodenticide market. The three chocolate dollops, approximately 1 g each, were also in each bait station in an effort to detect rodent activity (Figure 2) when mice did not feed on either of the toxic baits. While this does not reflect routine baiting practices by pest management professionals, adding chocolate spread bait allowed us to determine if the lack of feeding on the toxic baits was due to mouse elimination or due to mice’s lack of interests in the toxic baits.

In addition to placing rodenticide bait stations, an inspection of each apartment that had feeding activity was conducted and sanitation conditions and exclusion needs in each unit were recorded. The home sanitation level was rated as 1, 2 or 3, which represent “Good”, “Average”, or “Poor,” respectively [31]. The clutter level was rated as good, average, and cluttered (corresponding to the scale of 1–3 or above defined by the International OCD Foundation; http://www.hoardingconnectioncc.org/Hoarding_cir.pdf (accessed on 1 December 2020). This was documented and reported to the property management, following the inspection, to make repairs or work with residents to reduce conditions that were conducive for house mouse infestations.

On weeks 2, 4, 5, 7 and 9 following the initial installation of the rodenticide bait stations, each station was carefully inspected for bait consumption. If more than 10% feeding occurred on a rodenticide bait packet or chocolate dollop, or if any of the bait appeared to be damaged by human or environmental disturbance (e.g., water spills), the rodenticide bait packet was replaced with a new packet and the chocolate dollops were replaced after recording the weights, in the same manner previously described. If any bait stations were missing during an inspection, they were replaced with a new station with fresh baits. Consumption patterns of other pests (cockroaches, grain beetles, and ants) were analyzed using a hand lens to confirm house mouse feeding.

Following the week 4 visit, snap traps were installed in any apartment that had two subsequent visits without any feeding activity or evidence of house mice observed. Three TrapRite^®^ boxes (Anstar Products, Inc., Niles, IL, USA) with two Victor^®^ Easy Set^TM^ Mouse snap traps (Woodstream Corp., Lancaster, PA, USA) in each box were placed in locations in the same areas as the existing bait stations, but about six inches from the stations. Bait stations remained in place with the snap trap boxes. In each box, one snap trap was baited with a dollop of Hershey’s Chocolate Spread and the other was baited with a ~1 g dollop of Provoke^®^ Professional Gel (Bell Laboratories, Inc.) attractant. The addition of snap traps was to help further confirm if mice were eliminated after feeding activity on toxic baits had ceased. All equipment, including the snap trap boxes and bait stations were removed from all apartments after the week 8 visit.

A final inspection of all the treated apartments was conducted at week 10 following the initial installation of the rodenticide bait stations, during which time non-toxic baits and the chocolate spread in bait boxes that were installed in each apartment for one week to further verify apartments where mouse activity ceased after the treatment materials were removed. The method was the same as described for the initial building-wide inspection. Apartments were recorded as either “still active” or “not active” based on feeding activity as previously described. This information was given to the property managers at the conclusion of the rodent management program. Treatments conducted after the management program was completed were deployed as per the commercial pest control contracts in place.

### 2.4. Follow-up Mouse Inspections at 6 and 12 Months

Building-wide mouse activity was re-evaluated at six months and 12 months after the initial building-wide inspection. The methods deployed for each follow-up inspection were the same as those used to conduct the initial building-wide inspection. A summary of each building-wide inspection was provided to the property managers with additional specific recommendations for the contracted pest professionals, including: apartments with chronic mouse activity, apartments with activity not previously noted, and those apartments inspected but completely lacking any rodent activity.

### 2.5. Data Analysis

Chi-square tests were used to examine the associations between house mice presence with sanitation and clutter. Bait consumption occurrences between the two different non-toxic or two different toxic baits by mice were also examined using Chi-square tests. The infestation rates were calculated by dividing the total number of apartments with activity by the total number of apartments on each elevated area (e.g., floors 0–3 and floors 4+). Fisher’s exact test was conducted to compare the house mouse infestation rates of apartments located at lower and upper floors. All statistical analyses were conducted using SAS software (version 9.3; SAS Institute, Cary, NC, USA) [33].

## 3. Results

### 3.1. Initial Inspection and Resident Interview

A total of 170 residents (a participation rate of 38%) were present and agreed to answer the interview questions during the initial building-wide inspection. A total of 18 residents complained about mouse infestations, ten of which (56%) were not confirmed based on food bait or commercial blank rodent baits. These ten apartments were also found without mice during the six-month building-wide inspection. Among the 170 apartments whose residents were interviewed, 30 had existing house mouse activity based upon feeding in bait stations laid for a one-week period. Among those apartments with mouse activity, only eight (27%) residents noticed the presence of mice.

Among the 414 accessed apartments, 85% had sanitation and clutter levels rated as average to good while 15% contained poor sanitation or excessive clutter conditions. None of the vacant units were found to have house mouse activity. Only two apartments had both poor sanitation and excessive clutter. Poor sanitation or excessive clutter was not related to the presence of house mice when considering sanitation only (χ^2^ = 1.2, df = 1, *p* = 0.38) or clutter only (χ^2^ = 1.0, df = 1, *p* = 0.30).

Overall, 73% of the interviewed residents were unaware of a house mouse presence in their apartments as determined by house mouse feeding activity, while only 64% of the surveyed residents were concerned about mouse infestations. Table 1 summarizes the residents’ perceptions about mice. The majority of respondents (80%) agreed that mice are important vectors of disease, disagreed (52%) that mice are easy to eliminate and agreed (54%) that mice prefer dirty apartments. For residents who answered that they have used “do-it-yourself” products for controlling mice, the top three methods used were (from most to least common) glue boards (45%), rodenticides (21%), and snap traps (17%). Other “do-it-yourself” methods (e.g., electronic repelling devices, decluttering and cleaning) represented 17% of the responses.

### 3.2. House Mouse Management Program Effectiveness

From the initial building-wide inspections, 19 apartments (8.6%) were found to have house mouse activity at Trenton and 49 apartments (26.9%) were found to have house mouse feeding activity at Linden, for a total of 68 apartments with infestations. In addition, feeding activity was found in the trash compactor room and trash chute closets on floors 2–4 at the Trenton location.

At ten weeks after the installation of toxic baits, the percentage of treated apartments with feeding activity decreased from 89% to 39% at the Trenton location and from 61% to 2% at the Linden location (Figure 3). In addition, feeding in the trash chute closets on floors 2–4 at Trenton ceased. Bait consumption continued in the trash compactor room on the first floor. Overall, the number of infested apartments decreased by 63% and 98% in Trenton and Linden, respectively. The mean reduction rate was 87%. Among the 68 apartments with snap traps installed, a total of seven mice were captured in three apartments. Five mice were captured in one apartment and one mouse was captured in each of the other two apartments.

At six months, six and three apartments that had feeding during the initial inspection continued to have feeding activity at Trenton and Linden, respectively. Additionally, 12 and two new infestations were identified after the initial building-wide inspection at Trenton and Linden, respectively. As a result, the infestation rate increased slightly from 8.4% to 8.5% at Trenton and decreased significantly from 26.9% to 3.2% at Linden (Figure 4).

At 12 months, only three infested apartments identified at the beginning of study still had activities. They were located in Trenton. Additionally, 19 and three new infestations were identified in Trenton and Linden, respectively. The final infestation rate was 11.4% at Trenton and 1.7% at Linden. The number of apartments with house mouse activity increased by 26% in Trenton while decreased by 94% in Linden (Figure 4).

The recommendations to building management included 55 exclusion needs and eight sanitation improvements. At the Trenton location, 12 of 26 exclusion recommendations and two of the six sanitation improvement recommendations were completed, and the degree of compliance recorded. Of particular importance for house mouse management was the condition of the ground level compactor room, which collected all of the apartments’ trash in both buildings. One of the four exclusion and one of the three sanitation recommendations was completed at the trash compactor room at the Trenton location. At Linden, a total of 29 exclusion recommendations and two sanitation recommendations were reported to management. Twenty-six of the 29 exclusion recommendations and all of the sanitation recommendations were completed. In total, the Trenton and Linden locations had 44% (14 out of 32 recommendations) and 90% (28 out of 32 recommendations completed) compliance rates, respectively, for completing the exclusion and sanitation recommendations combined.

### 3.3. Detection of Mouse Infestations by Chocolate Spread and Two Non-Toxic Baits

During the initial building-wide inspection, 68 out of the total 414 apartments inspected were identified as having house mouse activity through a combination of resident interview and observed feeding upon baits in bait stations installed for a one-week period. Detex^®^, Liphatech^®^ Rat & Mouse Attractant, and chocolate spread, detected 17, 38 and 97% of the infestations, respectively (Figure 5). If using any of these two commercial baits alone for monitoring mice activities at least 62% of the infestations would have been missed. During the 12 month building-wide inspection, 39 apartments with house mouse activity were identified. Chocolate spread, Detex^®^, and Liphatech^®^ Rat & Mouse Attractant detected 77, 18 and 21% of the infestations, respectively. Chocolate spread detected an average of 89% of the infestations based on the three building-wide inspections.

When all non-toxic bait feeding data are combined, mice fed upon chocolate spread more often than either non-toxic bait (χ^2^ = 322.0, df = 1, *p* < 0.0001). Liphatech^®^ Rat & Mouse Attractant was consumed more often than Detex^®^ (χ^2^ = 335.3, df = 1, *p* < 0.0001). These data demonstrate that for the house mice located in these apartment buildings, chocolate spread was more effective than the two commercial non-toxic food baits in detecting house mouse activity. Liphatech^®^ Rat & Mouse Attractant bait was more effective than Detex^®^ for detecting house mouse activity.

Table 2 shows the feeding activity for chocolate spread compared to either commercially available non-toxic bait. In 70% of the cases where chocolate was consumed, neither Detex^®^ nor Liphatech^®^ Rat & Mouse Attractant was fed upon. Thus, when mice were offered a choice between the three food types used in this study, mice most frequently consumed chocolate spread.

### 3.4. Consumption among Chocolate Spread and the Two Toxic Baits

During the week 2, 4, 5, 7, and 9 visits after the management program had been initiated, a total of 154 feeding activities were recorded in bait stations. The feeding was grouped as chocolate only, toxic bait only, and toxic bait plus chocolate. Mice fed upon the chocolate spread significantly more often than the commercial toxic rodent baits (χ^2^ = 11.9, df = 1, *p* = 0.001). Of the 154 feeding activities recorded during the treatment period, chocolate spread, Contrac^®^ and FirstStrike^®^ were consumed 94%, 38%, and 59% of the time. Mice fed on the toxic baits without feeding on chocolate spread occurred in nine of the 154 feeding activities recorded.

Among 150 instances where feeding of either Contrac^®^ or FirstStrike^®^ or both occurred in bait stations, Contrac^®^ and FirstStrike^®^ were consumed 39% and 61% of the time, respectively. Contrac^®^ was consumed less frequently than FirstStrike^®^ (χ^2^ = 113.0, df = 1, *p* < 0.0001). The mean consumption per infested apartment containing Contrac^®^ and FirstStrike^®^ was 1.8 ± 0.3 and 2.0 ± 0.2 g, respectively.

Among 46 feeding activities recorded at week 4, 24% of the apartments with bait feeding was recorded from chocolate only. By week 11, 55% of the 22 feeding activities had only chocolate spread consumed. The preference of chocolate spread by mice compared to the toxic baits increased significantly (χ^2^ = 6.2, df = 1, *p* = 0.01).

### 3.5. Rates of Apartments with House Mouse Activity between Lower and Upper Floors

Table 3 summarizes the mouse activity occurring in apartments on the lower three floors of each building versus those on floors above the first three floors. Fisher’s exact test showed significant difference in the house mouse activity rates between upper floors and lower floors at each observation period and each site where enough samples are available. The odds ratios showed that apartments on the lower floors were between 5.7–18.2 times more likely to have house mouse activity than apartments on the upper floors.

## 4. Discussion

This study exemplifies the importance of a proactive rodent management program based on IPM necessary to succeed in long-term building-wide elimination of house mouse infestations in MFDs. A reactionary approach, relying on residential complaints to identify infestations and utilizing single tactics such as laying glue boards or poison baits, is insufficient and confirms similar findings regarding rodents [11], bed bugs [25,34] and cockroaches [32]. Although only two buildings were evaluated for house mouse management in this study, the findings presented here demonstrate that using effective monitoring methods, multiple efficacious control tactics and continued follow-up, is necessary to ensure sustainable mouse control in MFDs. For example, 73% of the interviewed residents were unaware of house mouse presence in their apartments as determined by house mouse feeding activity, demonstrating that residents’ reporting is insufficient for identifying apartments with house mouse activity. While 9–10% of the apartments were inaccessible, the proportion of the building monitored during the building-wide inspections allowed the inspectors to have a good understanding of where mice were present in order to implement the management program. House mouse management in high-rise MFDs must include this type of building-wide monitoring of both the primary areas of mouse activity and the routes of travel within the structure of MFDs over time.

At the two buildings included in this study, mice were found to be more prevalent on the lower three floors. The presence of mice inside of buildings is closely associated with poor exclusion and improper trash management and/or sanitation programs [13]. Trash compactors were located on the lower floor of both buildings researched. In the Trenton building, the trash compactor room had significant exclusion and sanitation deficiencies. This room consistently had the highest level of activity along with the lower floor trash chute closets. Where exclusion and sanitation recommendations were followed though, decreases in house mice feeding were observed. For example, exclusion recommendations were followed in the trash chute closets on floors 2–4 at Trenton and feeding recorded in these locations ceased without return. It is therefore reasonable to assume that the house mouse activity observed was caused by individuals from outside populations invading the building and exclusion was necessary for effectively reducing interior populations.

Interestingly, when considering the sanitation and clutter levels of individual apartments, there was no relationship between sanitation or clutter with house mouse activity in individual apartments. This could be due to a high level of house mouse movement throughout the buildings as a whole. This possibility would be supported by the frequency of occurrences where feeding would cease, only to return weeks later. We found five and six occurrences of where house mice feeding ceased and then returned during the management treatment phase of the study at the Trenton and Linden locations, respectively. If house mice are able to roam freely throughout a high-rise MFD, then it would make sense that sanitation or clutter levels within individual apartments would not be as critical a factor as building-wide trash management or exclusion. This would need to be formally tested. Proper building-wide exclusion and trash management should be implemented as part of an overall rodent management system rather than apartment by apartment [35,36].

The proportion of apartments that had feeding on only chocolate spread increased over the first three months of the study during the management program. Food preference in Muridae, including house mice, is largely dependent on the environment in which the mouse is living [37,38]. While this study did not evaluate the occurrence of chocolate food sources in these buildings, it would be logical that these house mouse populations experienced chocolate before as wild house mice typically exhibit food neophobia with novel food sources [39,40].

The results of this study show that, in these MFD environments and when offered three bait choices within each bait station, chocolate spread was more effective in identifying where house mouse activity was occurring. At least 62% of apartments with house mouse activity would not have been detected using the two commercial non-toxic baits examined in this study if offered a choice between commercial non-toxic baits and chocolate spread. Moreover, and in addition to wide-scale observations of a chocolate preference by mice among tenured pest professionals that have trapped mice for upwards of a century or more, formal research [41,42] has shown mice to possess a high preference for chocolate. We do not believe, therefore, that it was a rejection of the baits with the rodenticides so much as it was a high preference of chocolate over many other food choices when mice unexpectedly encounter chocolate during feeding forays.

Chocolate spread is an excellent bait choice for monitoring house mouse populations for several reasons: it poses no nut allergen risk to building occupants, it allows for quick and easy consumption level measurements, it remains palatable for a relatively long period in field conditions, and it is not easily disturbed with bait station movement as it remains attached to the station base. Additionally, chocolate spread, is readily available and is easy to apply. However, chocolate spread also attracted more non-target arthropods than the commercial baits did during this study. More research on the palatability and practical application of monitoring baits, chocolate spread, and several other food baits can lead to a better bait matrix for robust monitoring of all individuals in a population.

The need to investigate toxic bait preference by house mice is exemplified by the results of this study. The average consumption per apartment over a ten-week period among apartments with house mouse activity was 1.8 and 2.0 g for Contrac^®^ and FirstStrike^®^, respectively, which is much less than the average mean daily food consumption rate per mouse previously recorded as 3 g. [43]. The published LD_50_ of bromadiolone and difethialone is 1.75 mg/kg [44] and 0.47 mg/kg [45], respectively. With an average adult house mouse weight of 26.5 g, the feeding rate found during this study is comparable to ingesting 0.049 mg/kg and 0.027 mg/kg per day of bromadiolone and difethialone, respectively, which is much less than the LD_50_ levels for each rodenticide. Assuming multiple mice existed in each infested apartment, the consumption per mouse would be lower than the above calculated numbers. This could potentially amplify rodenticide resistance in populations that do not feed on a lethal dose; however, this study did not measure resistance.

The low level of rodenticide bait feeding may be due to the random sporadic feeding with more intense feeding in a few locations that house mice exhibit [46]. Alternatively, it can be due to the lack of palatability of the rodenticide baits. Regardless, the small consumption rates create a challenge for using currently available commercial baits to effectively attract or manage house mice. The feeding reduction found during the ten-week treatment period by researchers exemplifies the effectiveness of commercial rodenticide baits; however, the low (63%) reduction in apartments with house mouse activity after ten weeks of continuous rodenticide baiting in Trenton demonstrates that relying on rodenticides may not achieve full elimination. Incorporating exclusion to prevent continual mouse entry from outside and the use of a combination of several preventive control methods will likely achieve better control results.

The residents’ disposition towards house mouse infestations and management show how education is sorely needed to help facilitate residents partnering with the property managers and contracted pest professionals. Over a third (36%) of residents questioned were not concerned about mice, while relatively high levels of house mouse activity were found in both buildings. The most common “do-it-yourself” method found during the interviews was the use of glue boards, which are not recognized as an effective tool for controlling mice [47,48]. The findings demonstrate the dire need for information transfer regarding the importance of true IPM and the essential three-way partnerships among residents, building management staff, and the pest control provider. A rodent IPM program would include exclusion practices and a robust sanitation program, both of which was not fully implemented in this study. When information transfer is in place between the parties of a three-way partnership, exclusion and sanitation recommendations can be acted upon to assist in management tactics.

## 5. Conclusions

There are several conclusions and recommendations that are apparent from this research project. First, a building-wide survey using effective methods is necessary to identify areas of rodent activity. Resident surveys were found to be unreliable in locating house mouse activity, as was monitoring using the commercial non-toxic baits tested. Chocolate spread was much more effective than the two commercial blank baits for detecting house mouse activity, and it detected an average of 89% of the infestations. A building-wide mouse control program using baits as primary method of control resulted in 87% reduction in mouse activity after three months in two buildings. However, long-term (i.e., sustainable) house mouse control requires employing multiple control methods, including the use of several rodenticide baits with different characteristics to manage house mouse activity along with other critically important tactics such as exclusion and trash management. Finally, our results demonstrate that the IPM approach, including interior and exterior building envelope exclusion and proper solid waste management, is particularly essential for managing mice in the unique structural environments of large-scale multi-family dwellings as were studied here.

## Figures and Tables

**Figure 1 animals-11-00648-f001:**
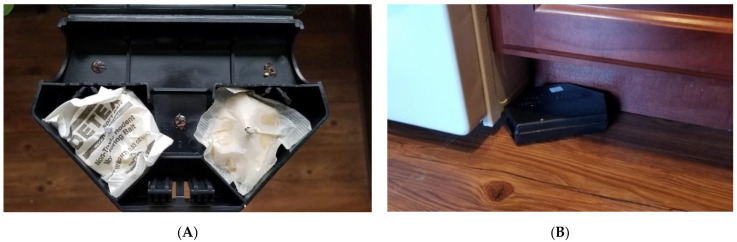
Protecta^®^ EVO^®^ Mouse bait stations for detecting house mouse activities. (**A**) An opened bait station showing non-toxic baits (Left: Detex^®^ soft bait; Right: Liphatech^®^ Rat & Mouse Attractant) and three dollops of chocolate spread. (**B**) A bait station placed beside the stove in a kitchen.

**Figure 2 animals-11-00648-f002:**
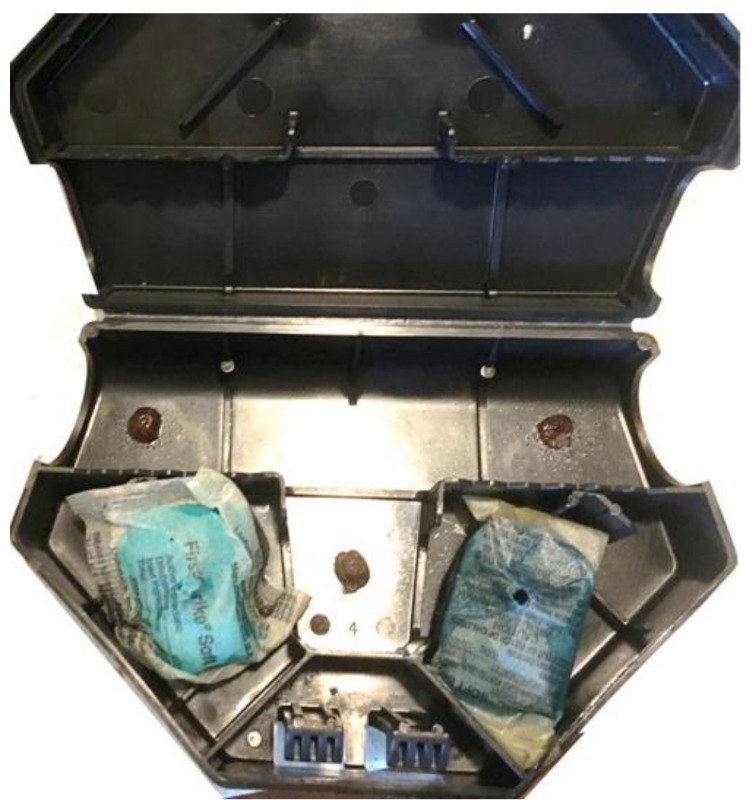
A Protecta^®^ EVO^®^ Mouse bait station with two toxic baits (Left: FirstStrike^®^ soft bait; Right: Contrac^®^ soft bait) and three dollops of chocolate spread.

**Figure 3 animals-11-00648-f003:**
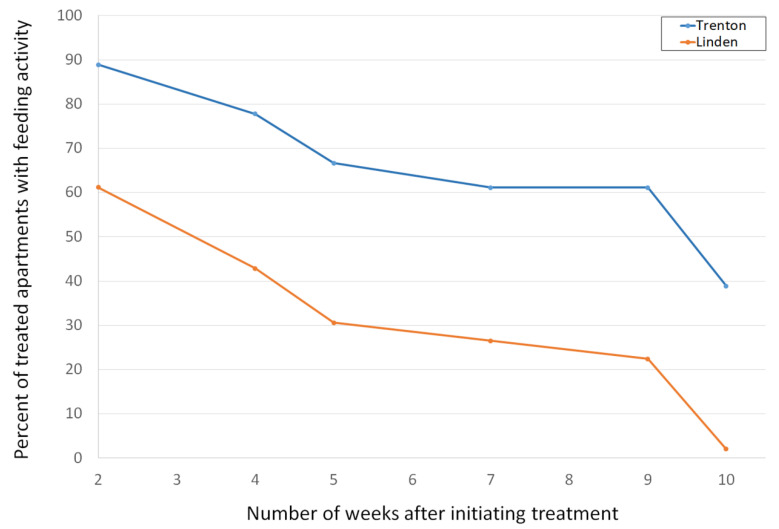
The dynamics of house mouse activity based on feeding activities on toxic baits in two high-rise apartment buildings after the implementation of a house mouse management program. The data are based on a total of 19 and 49 infested apartments at Trenton and Linden from January–April and February–June 2019, respectively.

**Figure 4 animals-11-00648-f004:**
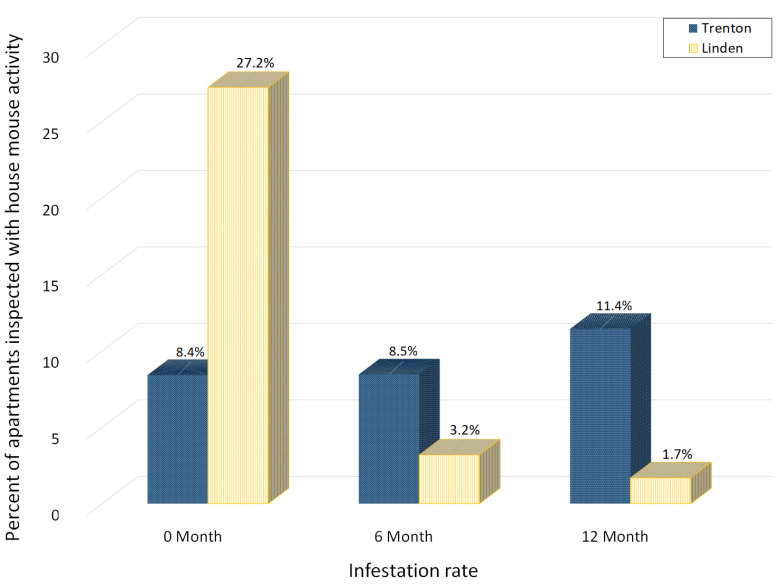
The percent of apartments with house mouse activity found during the building-wide inspections at 0, 6 and 12 months.

**Figure 5 animals-11-00648-f005:**
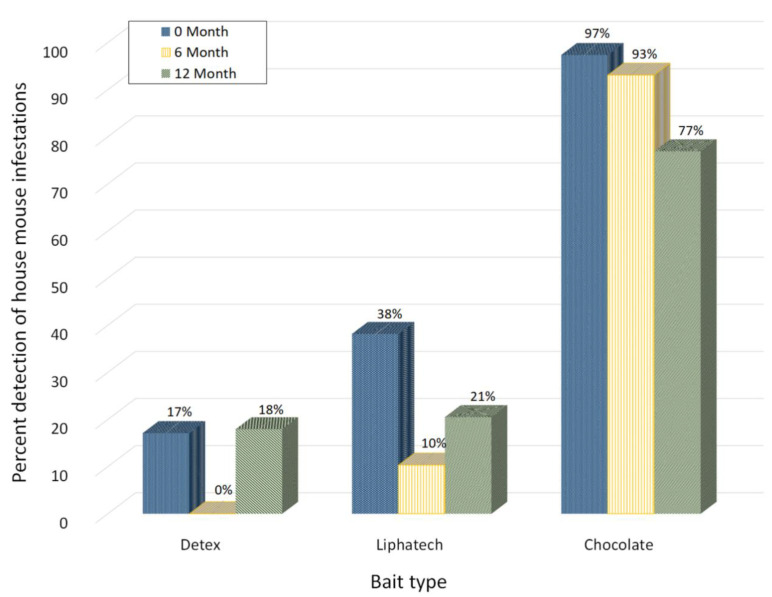
The percent of detection of house mouse activity by three different non-toxic baits during building-wide inspections.

**Table 1 animals-11-00648-t001:** Resident responses to questions regarding perceptions about mice regarding disease, management and conducive conditions (n = 145).

Question	Number (Percentage)		
	Agree	Disagree	Not Sure
Do mice cause disease?	136 (80%)	6 (4%)	28 (16%)
Are mice easy to eliminate?	40 (24%)	88 (52%)	40 (24%)
Do mice prefer dirty apartments?	91 (54%)	57 (34%)	19 (11%)

**Table 2 animals-11-00648-t002:** Comparison of feeding activity on chocolate spread and two commercial non-toxic baits during the 12 month study.

Commercial Baits Fed Upon When Chocolate Was Consumed	Number of Occurrences When Bait Options Were Fed Upon	Proportion of Occurrences When Bait Options Were Fed Upon
Neither bait had feeding activity	82	70%
One of the baits was fed upon	24	20%
Both baits were fed upon	12	10%
Total	118	100%

**Table 3 animals-11-00648-t003:** Distribution of mouse activity between the lower three floors and floors four and above.

Building	Period	Number of Apartments	% of Apartments Infested in Lower Floors	% of Apartments Infested in Upper Floors	Statistics	Odds Ratio
Trenton	Month 0	216	37.1%	3.1%	χ^2^ = 44.4, *p* < 0.0001	18.2×
Trenton	Month 6	213	28.9%	4.0%	χ^2^ = 25.1, *p* < 0.0001	9.8×
Trenton	Month 12	210	31.4%	7.4%	χ^2^ = 16.6, *p* = 0.0003	5.7×
Linden	Month 0	180	46.9%	23.0%	χ^2^ = 7.6, *p* = 0.0059	3.0×
Linden	Month 6	157	20.8%	0.0%	χ^2^ = 28.6, *p* < 0.0001	*
Linden	Month 12	178	0.0%	2.1%	*	*

* No data are available due to very small number of infested apartments.

## Data Availability

The data presented in this study are available on request from the corresponding author. The data are not publicly available due to the privacy requirements of the Institution Review Board for participants.

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
