# Peer review of "Monitoring and Controlling House Mouse, *Mus musculus domesticus*, Infestations in Low-Income Multi-Family Dwellings"

_animals, 2021, doi:10.3390/ani11030648_

Round 1
Reviewer 1 Report
The comments of the reviewers were well addresses.
Author Response
The comments of the reviewers were well addresses.
- Thank you for your time,
Reviewer 2 Report
The authors have addressed the points I raised.
Author Response
The authors have addressed the points I raised.
- Thank you for your time,
Reviewer 3 Report
Dear Authors,
The study you conducted is an interesting one.
As reviewer, I have some remarks/questions:
In the summary: add the location/world part where the research was conducted.
L29: delete the word ‘very’
L74/75: depletion of food
Introduction: I miss information on proper rodent management methods. Please elaborate and include information on IPM. I also miss basic information on mice and their behaviours and needs.
Are the residents renters or buyers? And what does this mean according to the law in relation to making a house rodent-proof?
L84-87: Why is the knowledge of the residents not questioned? Or the thing they undertake already or in the past in order to manage rodents inside?
L90: add: ‘in New Yersey’
L100: ‘a higher level’, what is higher? Can you specify this with a number?
L102: What is the distance between the buildings? How were they selected? Are the surroundings comparable?
L109: How do the surroundings look like? Can mice come from outside the buildings?
L113: What did the inspections look like? Be more precise
L116: this is a result and therefore should be moved to the results section
L119: What time period? How long were the bait stations used to monitor rodent activity? Maybe create a timeline to create a more clear overview of the research for the reader
L153: this is a result. Either rewrite, or move to the results section
L159: Avoid the use of ‘we’. This goes for the whole manuscript.
L161: Please add/include the list with recommendations.
L161: Where are the recommendations based on?
L163: how many bait stations were there used in total per building? Also needed for the results section, to give more insight in the total numbers
L189: when reading the M&M, the question raises why here is chosen to work with rodenticides instead of working according to IPM methods? Why is there not worked with snap-traps and implementing management methods in order to make each apartment and the building mouse-proof? (by making food, water, shelter, of nest facilities inaccessible)
L223: Where the bait boxes and traps removed? Or how many were left?
L266: Make the title of the table more clear. Add the n of the participants and the location. In such a manner that a reader can understand the table clearly, even without reading your manuscript.
Fig 3: Add year in the figure or description, and also add a start date
L291 - 297: please also give this information in a graph or table
L298: what sort of management recommendations? And how was it checked?
L311: 68 of a total of …
L314: Make sure the order of your numbers is mentioned in the same order in the figure (4). Now they are in a different order
L335: Doe this mean that rodents have the ability to detect rodenticides in baits? Please elaborate on this in the discussion
L337: Make this table more clear. A reader who has not read the manuscript, does not understand what the table is about (e.g. what is a ‘occurrence’? How many of total possibilities?)
L365: Make the title more clear. Describe that a lower floor and an upper floor is.
L368: add: based on IPM
L375: This is the first time I read this information (73%...). This should be in the results.
L378: Were these inaccessible apartments on the lower or top floors?
L394 onwards: Did you also study/check the routes mice could take within the building and apartments (so holes of >0.5cm from the one floor to the next for example in the fuse boxes or next to the heating pipes)
Have you considered the use of cameratraps to monitor instead of ‘feeding’ the mice using non-tox bait (=food)? And if so, why did you choose to use bait in this research and not a more IPM based method?
L429: You try to make a point here, but do not forget that the rodenticides used are multi-dose products. So the rodent needs to eat multiple times from it to die. This conclusion stated is therefore incorrect. Please adjust.
Discussion: I miss a part on rodenticide resistance. When rodent do eat a little form te rodenticides, but not enough to actually die, resistance can occur. Although the methods used in this study can be discussed, I at least expect a part on this in the discussion.
L442: add: ‘preventive’ before control methods
L442: add the word ‘of’
Discussion: Could the season have been of influence on the results of the research?
Maybe acknowledge the owners of the buildings, or the residents who participated.
Author Response
Thank you for your time,
Please find attached a document with individual responses to each comment as well as alterations to the text or other aspects of the manuscript as discussed.
Thank you again,

This manuscript is a resubmission of an earlier submission. The following is a list of the peer review reports and author responses from that submission.
Round 1
Reviewer 1 Report
Please see comments in the attached file

Reviewer 2 Report
I would like to thank the authors for submission of their manuscript on an important topic, as IPM requires more knowledge of the biology of pest species (in this case house mice) than we currently have.
My key remark concerns the scientific soundness of the study. It's a pity that a control building is missing in this study (although I can understand the reason for it as it will mean a lot of extra work), because it's also important to know what happens if no rodent control actions are taken at all. Also, the authors should be aware that in a true IPM approach, control is not the only aspect, but prevention (rodent proofing/habitat management etc.) is also very essential.
Moreover, as house mice are neophilic (they like anything that's new), I can imagine that they are more keen on chocolate drops than on those commercial baits. They may have experienced the latter in the past. Therefore, it would have been interesting to know what type of commercial baits were used by the property owners in the past. Moreover, like rats, house mice could experience the poisoned partner effect, as explained by N.W. Bond (1984). This could also influence their uptake of commercial baits and make chocolate more attractive. However, I underline the claim by the authors that more research on the palatability of commercial baits is necessary to increase their consumption by the pest species.
Best regards.
Reviewer 3 Report
This is a well set-up study on an interesting topic. Although IPM exists for a while, the implementation is not always well done. Studies investigating the application and methods of improvement are sorely needed. Here are my comments in order in the m/s.
Page 2 Introduction, 3rd paragraph: The definition of IPM is a little vague and does not include exclusion and/or sanitation, which are essential elements.
Page 3 2.2, 1st line. only 90-91% of apartments were accessed. Please add to the Discussion a comment on the effect this may have had on your findings.
2.2, 5th line 'found having mouse infestation ...' should be 'found to have mouse infestation...' (English)
Page 4 3rd paragraph: Please add the list of interview questions or questionnaire in an Appendix. This is very useful information for readers.
2.3, 2nd paragraph, last sentence: 'two interior doors'. It is not clear where these doors are - interior in each apartment or interior to the buildling? Please clarify.
Page 6 3.1, 1st paragraph, 5th line: Remove comma after 'apartments'.
3.1, 2nd paragraph, 3rd line: 'with' should be 'to' (English)
3.1 3rd paragraph, 5th line: 'top four methods'- only three methods are listed.
3.1, 3rd paragraph, 7th line - add a comma after 'devices'.
Page 7, last sentence of the paragraph starting 'At 12 months...': Trenton and Linden are suddenly listed in reverse order, which makes it confusing for readers. Please always refer to them in the same order.
Page 8, Fig. 4, x-axis: check spelling of 'Liphtech'
Page 10 4. Discussion, 4th paragraph, 'as critical of a factor': Please delete 'of'. (English)
Page 11 2nd paragraph, To what extent could the lower consumption of bait be due to competition with chocolate paste (in other apartments)?
P.S. I recommend the authors provide better references for the link between mice and public health on page 2 (Introduction, 3rd line references cited 1-5). One of the papers cited, actually shows mice were free of yersinia and salmonella. There are several recent papers demonstrating that mice in housing or in urban areas are carriers of infectious microorganisms.